# Preparation and Characterization of Paramagnetic Bis (8-Hydroxyquinoline) Manganese Crystals

**DOI:** 10.3390/ma13102379

**Published:** 2020-05-21

**Authors:** Feng Jiang, Jiawen Song, Mengqi Dong, Yinong Wang

**Affiliations:** The Mathematics and Physical Science Centre, Dalian University of Technology, 2 Dagong Road, Liaodongwan New District, Panjin 124221, China; jiangfeng@dlut.edu.cn (F.J.); Jiawensong@mail.dlut.edu.cn (J.S.); dllgdongmq@mail.dlut.edu.cn (M.D.)

**Keywords:** bis (8-hydroxyquinoline) manganese, nanorod, magnetic properties, DFT calculations

## Abstract

The magnetic properties of π-conjugated bis (8-hydroxyquinoline) manganese (Mnq_2_) crystals are investigated. Rod-shaped Mnq_2_ crystals are prepared by using the physical vapor deposition method. Field emission scanning electronic microscopy spectra show that the Mnq_2_ nanorods have perfect plane quadrangular ends. Energy dispersive spectrometer and X-ray photoelectron spectroscopy analysis demonstrates that the powders and nanorods are the same compound with a high purity. X-ray diffraction analysis shows the high crystal quality of the prepared Mnq_2_ nanorods. The magnetic measurement, using alternating gradient magnetometer and magnetic property measurement system superconducting quantum interference device vibrating sample magnetometer, indicates that the prepared Mnq_2_ nanorods show a paramagnetic property at room temperature. First-principles density functional theory (DFT) calculations are used to study the electronic structure and magnetic properties of the prepared Mnq_2_ crystals. DFT calculations show that the magnetic moment of the Mnq_2_ isolated molecule is 5 *μ_B_,* which mainly comes from the localized Mn 3*d* orbital. The energy difference between the antiferromagnetic and ferromagnetic states of the Mnq_2_ monoclinic cell is only 0.1 meV, which may explain the paramagnetic property observed in the prepared Mnq_2_ nanorods and also indicates the difficulty of preparing intrinsic ferromagnetic Mnq_2_ crystals.

## 1. Introduction

The π-conjugated 8-hydroxyquinoline metal complexes have been widely used in organic light-emitting devices and organic solar cells due to their excellent optoelectronic properties [1,2,3,4]. Moreover, the well-known tris (8-hydroxyquinoline) aluminum (Alq_3_) has been found to have a wide application in organic spintronic devices because of its large spin relaxation length [5,6]. Since the discovery of giant magnetoresistance in multilayered devices which use Alq_3_ as the spin transporting layer, there has been increasing interest in 8-hydroxyquinoline-based organic magnets owing to their special properties, for example, the conductivity matching and lattice matching between the organic transporting layer and the electrode in organic spintronic devices [6,7,8].

Compared with its organic, semiconducting and photoelectric properties, scarce research on the magnetic properties of 8-hydroxyquinoline small molecules has been reported, which is mainly because the commonly used 8-hydroxyquinoline small molecules, such as Alq_3_ and Gaq_3_, generally do not exhibit magnetic behavior [9,10]. In 2008, Baik et al., introduced ferromagnetism into Alq_3_ molecules by doping Co metal [11]. Our research group prepared several 8-hydroxyquinoline-based magnets by coevaporating organic materials and dopants [10,12,13]. For 8-hydroxyquinoline transition metal complexes Mq_x_ (M = Mn, Fe, Co, etc., and x = 2 or 3), the 3*d* orbital of the transition metals can provide the local magnetic moment. In 2011, Monzon et al. systematically researched the magnetic properties of several 8-hydroxyquinoline transition metal complexes existing in the form of powder or film and found that all of the researched 8-hydroxyquinoline transition metal complexes exhibited weak paramagnetic properties [14]. In powder or film state, the organic materials are amorphous and long-range disordered. It is difficult to form an effective ferromagnetic exchange coupling between the local magnetic moments. Due to the size and surface effects, the crystalline nanostructure may change the properties of the materials [15,16,17]. However, the effect of a crystalline structure on the magnetic properties of 8-hydroxyquinoline transition metal complexes has not been reported before. In this work, Mnq_2_ crystals were prepared by using a double-zone tubular furnace, and the electronic structure and magnetic properties of Mnq_2_ crystals were studied experimentally and theoretically.

## 2. Experimental Details

Mnq_2_ crystals were prepared by using the physical vapor deposition (PVD) method in an OTF-1200X-II dual-zone tubular furnace. Figure 1 shows the diagrammatic sketch of the tubular furnace, where region I is the sublimation zone and region II is the crystalline zone. Resistance wires surrounded the quartz tube at equal distance in order to obtain a uniform temperature distribution. The temperatures of the two zones were constantly monitored by using a thermocouple, and the temperature upgrade speeds were set to be 10 °C/min. The temperatures of the sublimation and the crystalline zones were 425 °C and 275 °C, respectively. The crystal growth time was 300 min, then the two zones were cooled down to room temperature naturally. The flow rate of Ar carrier gas (99.999%) was 0.35 L/min. The Ar gas plays the role of carrier gas and protective gas from the air and moisture. The Mnq_2_ powders (99%, purchased from Nichem company, Taiwan, Taiwan) were placed in the sublimation zone, and the Si (100) substrate was placed in the crystalline zone. The Si substrate was cleaned by ultrasonic cleaning in deionized water and absolute alcohol for 15 min in turn, then it was degreased in acetone. Before the preparation, the tubular furnace was washed with Ar gas three times to remove the air and moisture in case the Mnq_2_ molecules decomposed into 8-hydroxyquinoline at high temperature [18,19]. The surface morphology of the samples was characterized by field emission scanning electronic microscopy (FESEM nona nano450, FEI company, Hillsboro, OR, USA). The chemical composition of the powders and the crystals was investigated using an energy dispersive spectrometer (EDS, ATMETEK, Inc., Berwyn, PA, USA) and X-ray photoelectron spectroscopy (XPS ESCALAB 250xi, Thermo Fisher Scientific, Waltham, MA, USA). The structure of Mnq_2_ crystals was investigated by using X-ray diffraction (XRD, RigakuUltima IV X-ray diffractometer with Cu kα1 radiation, Rigaku Corporation, Tokyo, Japan). The room-temperature magnetic properties of the Mnq_2_ powders and crystals were measured using an alternating gradient magnetometer (AGM, MicroMag 2900, Lake Shore Cryotronics Inc., Westerville, OH, USA). The magnetic properties at different temperatures were measured using magnetic property measurement system superconducting quantum interference device vibrating sample magnetometer (MPMS SQUID VSM, Quantum Design company, Santiago, CA, USA).

## 3. Computational Methods

The calculations were performed using the CASTEP code. Generalized gradient approximation applied by Perdew−Burke−Ernzerhof [20] was adopted to estimate the exchange-correlation function. The cut-off energy was set at 450 eV and the valence configuration of Mn was 3*d*^5^4*s*^2^. The lattice parameters used for the calculation of the monoclinic cell which contains two Mnq_2_ molecules are *a* = 10.930 Å, *b* = 4.885 Å, *c* = 15.184 Å, *α* = 90.000, *β* = 121.210, *γ* = 90.000 [21]. For the Brillouin zone sampling, 1 × 1 × 1 Monkhorst−Pack [22] *k* mesh was used for the isolated Mnq_2_ molecule while 1 × 3 × 1 Monkhorst−Pack *k* mesh was used for the Mnq_2_ monoclinic cell. The structure relaxation was fully carried out until the forces on each ion were less than 0.01 eV/Å.

## 4. Results and Discussion

Figure 2 shows the SEM images of the prepared Mnq_2_ crystals. Regular rod-shaped Mnq_2_ crystals with random distribution could be observed from the SEM images. Most of the prepared Mnq_2_ nanorods had perfect plane quadrangular ends, as shown in Figure 2a,b. The end size of the Mnq_2_ nanorods was dozens of microns, and most of their lengths exceeded 200 microns. There were no obvious defects on the surface of the Mnq_2_ nanorods as shown in the SEM images, implying high crystal quality of the prepared Mnq_2_ nanorods. The excellent crystal quality might be caused by the low crystal growth rate, the pure growth conditions and the weak perturbation [23]. As shown in Figure 2c,d, there were small numbers of crystalline bulks on the surface of the Si substrate. The mechanism of crystal growth might be that the Mnq_2_ molecules carried by Ar gas aggregated to form a nucleation around the residual impurity or the root of the formed Mnq_2_ nanorods. As the process continued, the strong π-π interaction between pairs of quinoline ligands of the depositive Mnq_2_ and the gaseous Mnq_2_ molecules led to the formation and growth of Mnq_2_ nanorods along a special direction [24].

Figure 3a shows the EDS spectra of the Mnq_2_ powders and prepared Mnq_2_ nanorods. The peaks which stood for C, N, O, Mn elements existed in both the powders’ and nanorods’ spectra. Beyond that, there were no peaks appearing in either of the two spectra, showing that the Mnq_2_ powders and prepared Mnq_2_ nanorods had the same chemical composition, and there were almost no impurities in either the powders or the nanorods. Figure 3b is the XPS spectra of survey for the Mnq_2_ powders and nanorods. From the survey spectra in Figure 3b, the peak positions which stood for the C 1*s*, N 1*s*, O 1*s* and Mn 2*p* core levels located at 285, 399, 532 and 641 eV, respectively, could be found in both the powders and the nanorods. The EDS and XPS measurements demonstrated that the powders and nanorods were the same compound and both were of good purity.

Figure 4a shows the XRD diffractogram of the Mnq_2_ powders and prepared Mnq_2_ nanorods. The number of diffraction peaks in the XRD patterns of Mnq_2_ powders was large, and the baseline was not straight. Compared with the XRD spectra of the powders, the baseline of the Mnq_2_ nanorods was almost horizontal, and the full width at half maximum of the diffraction peak was smaller, indicating that the prepared Mnq_2_ nanorods had a high crystal quality. Four sharp diffraction peaks appeared at 2θ = 7.7, 8.3, 29.2 and 32.9˚. These four diffraction peaks also existed in the XRD diffractogram of the Mnq_2_ powders. Beyond these four diffraction peaks, there were a number of diffraction peaks at different angles in the XRD diffractogram of the Mnq_2_ powders and the XRD diffractogram of Mnq_2_ powders was broad. This result showed that there existed the same crystalline phase in the Mnq_2_ powders as in the prepared Mnq_2_ crystals. Beyond that, there existed other crystalline phase(s) with a quantity of amorphous powders in the Mnq_2_ powders [25]. The XRD diffractogram indicated that the growth process increased the degree of crystallinity of Mnq_2_.

Figure 4b shows the magnetization curves of Mnq_2_ powders and nanorods with the same weight (1.36 mg) measured by AGM at room temperature. The Mnq_2_ powders exhibited paramagnetic properties at room temperature, which was consistent with the previous report [14]. The magnetization was 0.18 emu/g at 14,000 Oe. Transition from Mnq_2_ powders to Mnq_2_ nanorods did not change the magnetic property of the sample. The Mnq_2_ nanorods still exhibited a paramagnetic property. The difference is that the magnetization of Mnq_2_ nanorods was much larger than that of Mnq_2_ powders with the same weight, which was 0.53 emu/g at 14,000 Oe. As shown in the SEM image and XRD diffractogram, the Mnq_2_ nanorods had a higher degree of crystallinity than the Mnq_2_ powders. The larger crystalline grain size of Mnq_2_ nanorods reduced the degree of magnetic moment confusion on the surface, so the loss of magnetic moment caused by surface magnetic moment was reduced. On the other hand, the higher degree of crystallinity for the Mnq_2_ nanorods may have brought about a stronger crystalline field, making the magnetic moments more orderly. These factors increased the magnetization of Mnq_2_ nanorods.

Figure 5 shows the magnetization curves of the prepared Mnq_2_ nanorods at different temperatures measured by MPMS SQUID VSM. The weight of the test sample was 11.03 mg. The magnetization of the nanorods at 20,000 Oe was 0.81 emu/g at 300 K. The Curie constant was calculated to be 4.17 emu K/mol. The Curie constant was very close to the value given by one Mn (II) ion per mole of a compound (about 4.38 emu K/mol). The intensity of magnetization increased as temperature decreased, as shown in Figure 5. The reciprocal of magnetic susceptibility for the nanorods as a function of temperature is shown in the inset of Figure 5. The susceptibility was almost linear with temperature. This conformed to the rule of paramagnetic materials.

First-principles density functional theory (DFT) calculations were used to study the electronic structure and magnetic properties of the sample. There is no structure information for the Mnq_2_ molecule, so we used the structure of Cuq_2_ molecule [21], replacing the intermediate Cu atom with a Mn atom, as shown in Figure 6a. The Mnq_2_ isolated molecule was placed in a sufficiently large cell (a vacuum space of 20 Å) to avoid interaction with images reproduced by periodic boundary conditions [26]. Figure 6a shows the structure of Mnq_2_ isolated molecules after full relaxation. Unlike the Alq_3_ molecule which had different Al-O and Al-N bond lengths [24], the Mnq_2_ molecule had planar structure, and the two quinoline ligands were symmetrical about the middle Mn atom. The two Mn-O bonds and the two Mn-N bonds had equal bond lengths, which were 1.886 and 1.977 Å, respectively. The different bond lengths of Mn-O and Mn-N bonds implied that the central Mn atom bonded more tightly with the adjacent O atoms than the N atoms. This result was in good accordance with the previous reports calculated by using the Vienna ab initio simulation package [27].

If a system is magnetic, the local magnetic moment ought to exist. The magnetic properties of the Mnq_2_ isolated molecule were calculated after full relaxation. Figure 6b shows the total density of states (DOS) of the Mnq_2_ isolated molecule. The total DOS exhibited obvious exchange splitting between the majority and minority spin channels around the Fermi level. The total magnetic moment was 5 *μ_B_*, which was mainly localized on the Mn atom (4.94 *μ_B_*). There was a transfer of 1.78 electrons from the Mn 4*s* orbital to the quinolone ligands, leaving the 4*s* orbital with 0.22 electrons, while the 3*d* orbital had 5.04 electrons. Figure 6c shows the partial DOS (PDOS) of Mn 3*d* orbital in the isolated Mnq_2_ molecule. Most of the Mn 3*d* electrons occupied spin-up state, which conformed to Hund’s rules. The 3*d* orbital of the corresponding Mn ion was partially occupied. The different electron occupations of majority and minority spins led to a net local magnetic moment. The adjacent nonmetal atoms (C, N and O atoms) gave little contribution to the magnetic moments, due to the nearly closed shell of the nonmetal atoms and the somewhat delocalized characteristics of the corresponding ions [26].

To study the magnetic interaction between the local magnetic moments, the Mnq_2_ monoclinic cell was studied by DFT calculation. After full relaxation, the lattice parameters of the Mnq_2_ monoclinic cell which contained two Mnq_2_ molecules were a = 11.425 Å, b = 5.433 Å, c = 16.358 Å, α = 90.000, β = 118.55711, γ = 90.000. The energy differences between antiferromagnetic (AFM) and ferromagnetic (FM) states of the Mnq_2_ cell were calculated after full relaxation. The energy differences (ΔE) and the magnetic coupling of the four different initial spin configurations on Mn atoms of the Mnq_2_ monoclinic cell are shown in Table 1. As shown in Table 1, the FM states had lower energy than the AFM states, implying that the FM state was the ground state magnetic order. ΔE_L_ was the energy difference between the current state and the ground state. However, the energy difference between the FM and AFM states was only 0.1 meV, which was much smaller than the room-temperature thermal energy (about 26 meV) [28]. These calculation results show that it is difficult to prepare intrinsic ferromagnetic Mnq_2_ crystals, and these results also explain the paramagnetism of the prepared Mnq_2_ nanorods. The distance between the two Mn atoms changed from 7.975 to 8.618 Å after full relaxation. The long distance made it difficult to form an effective ferromagnetic coupling between the two local magnetic moments.

## 5. Conclusions

Mnq_2_ nanorods are prepared by using the PVD method. The SEM and XRD analyses indicate the high crystal quality of the prepared samples. The AGM test shows that Mnq_2_ nanorods exhibit paramagnetic behavior, and the magnetic field response of the Mnq_2_ nanorods is much stronger than that of the powders. First-principles DFT calculations show that the Mnq_2_ isolated molecule has planar structure, and the magnetic moment is 5 *μ_B_*, which mainly localizes on the 3*d* orbital of the Mn atom. The energy difference between FM and AFM states was only about 0.1 meV, indicating that it was difficult to prepare intrinsic ferromagnetic Mnq_2_ crystals, and explaining the experimental paramagnetic behavior of the prepared Mnq_2_ nanorods.

## Figures and Tables

**Figure 1 materials-13-02379-f001:**
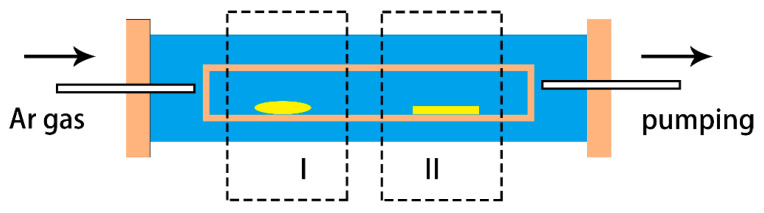
A diagrammatic sketch of the dual-zone tubular furnace.

**Figure 2 materials-13-02379-f002:**
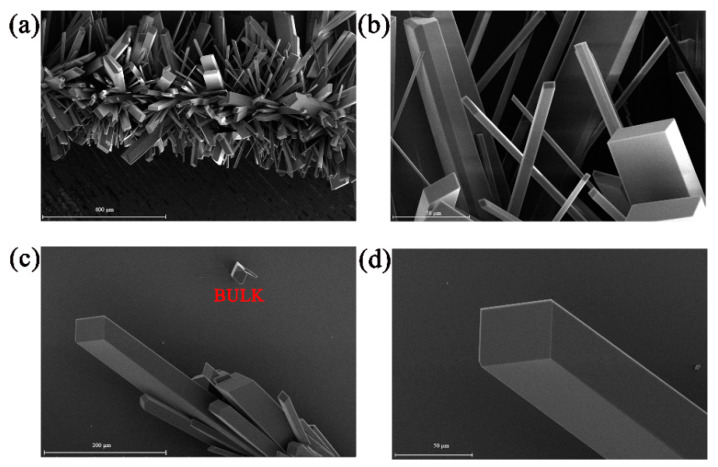
The SEM micrographs of the prepared Mnq_2_ crystals. Rod-shaped microstructures with (**a**) 400 μm and (**b**) 50 μm scale. Rod-shaped and nubby microstructures with (**c**) 200 μm and (**d**) 50 μm scale.

**Figure 3 materials-13-02379-f003:**
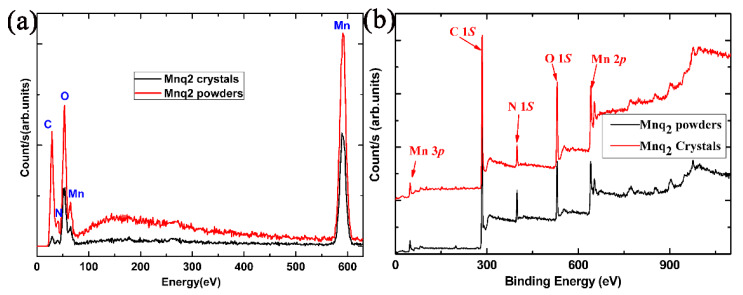
(**a**)The EDS spectra of the Mnq_2_ powders and prepared Mnq_2_ nanorods. (**b**) The XPS survey spectra for the Mnq_2_ powders and nanorods.

**Figure 4 materials-13-02379-f004:**
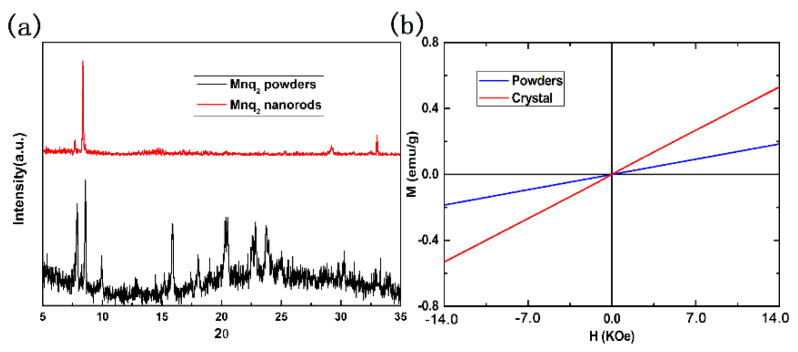
(**a**) XRD of Mnq_2_ powders and nanorods. (**b**) The M-H curves of Mnq_2_ powders and nanorods measured by alternating gradient magnetometer (AGM) at room temperature.

**Figure 5 materials-13-02379-f005:**
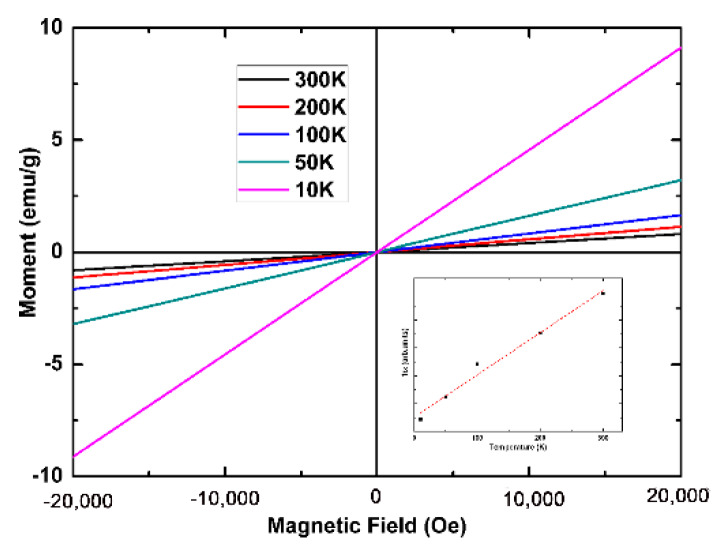
The M-H curves of Mnq_2_ nanorods measured by SQUID VSM at different temperatures, with the plot of susceptibility vs. temperature in the inset.

**Figure 6 materials-13-02379-f006:**
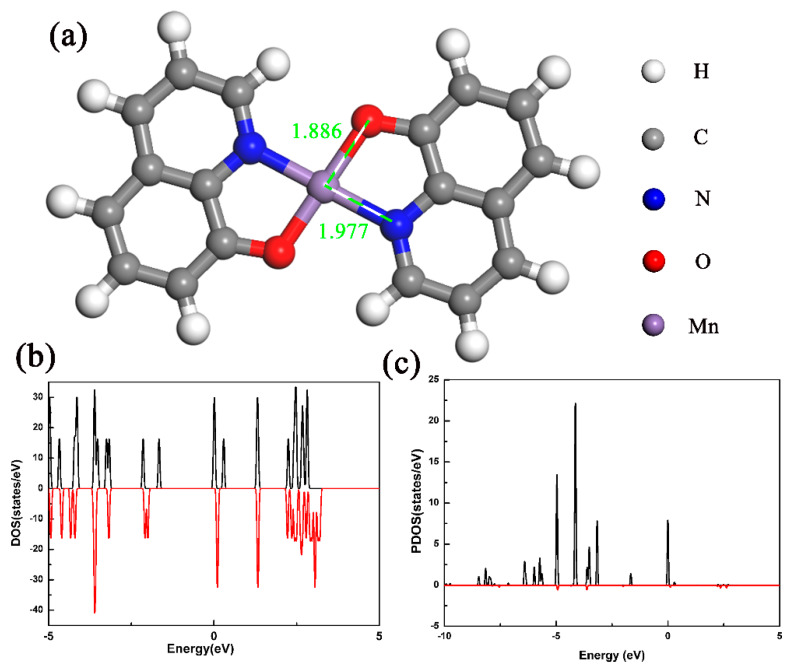
(**a**) The molecular structure of Mnq_2_ molecule after full relaxation. (**b**)The total DOS of the Mnq_2_ isolated molecule. (**c**) The PDOS of the Mn 3*d* orbital in the Mnq_2_ isolated molecule.

**Table 1 materials-13-02379-t001:** The energy difference between antiferromagnetic (AFM) and ferromagnetic (FM) states, and the corresponding magnetic coupling of different initial spin configurations on Mn atoms of the Mnq_2_ monoclinic cell.

Initial Direction		∆E_L_ (meV)	Magnetic Coupling
Mn_1_	Mn_2_		
↑	↑	0	FM
↑	↓	0.1	AFM
↓	↑	0.1	AFM
↓	↓	0	FM

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
