# Peer review of "Preparation and Characterization of Paramagnetic Bis (8-Hydroxyquinoline) Manganese Crystals"

_materials, 2020, doi:10.3390/ma13102379_

Round 1
Reviewer 1 Report
The authors reported synthesis, structure and magnetic properties of π-conjugated bis(8-hydroxyquinoline) manganese (Mnq2). However, the magnetic measurements are very limited. Major changes must be implemented:
Why the magnetization is not calculated in emu/g? Preparing samples in the same weight are quite odd method. The magnetic measurements should expanded. Measurements in higher fields, for M vs. H and temperature dependence of magnetization (M vs. T) need to be done. Both measurements should be carefully analyzed. If you know the formula and weight, you can easily calculate the values for paramagnetic case (Curie constant, M vs. H curve) and compare with experimental values. For simplicity you can use the same assumption as for DFT calculations. After rewriting the magnetic experimental part you can compare yours calculations.
Minor comments:
The first sentence in abstract is inadequate. The abstract should not survey the field but specifically tell details about your study and the outcomes/conclusions
Based on the comments mentioned above I recommend a major revision.
Reviewer 2 Report
High quality crystals of Mnq2 have been prepared. The magnetic study of these crystals has been performed. However, I am not convinced by the comparative study with the powder. In my opinion, the powder needs much more analysis to determine its composition. Beside this fact, there is no structure of the nanorods and why must we assume that they have the same structure as the Cuq2. Can we see both XRPD diffractograms, of Mnq2 and of Cuq2 to check that they are, really, isostructural. Otherwise the clue could be to prepare the crystals with Zn which normally works well.
The XRPD diffractogram Figure 3 (and not spectra) shows that the powder could be a mixture of different structures with different proportions of Mn, so it is not rigorous to compare Mnq2 powder and nanorods.
Round 2
Reviewer 1 Report
Unfortunately, I do not feel convinced by the analysis of the magnetic data. I understand, that the present situation related to the SARS-CoV-2 virus causes significant difficulties. However, the presented data shows some kind of magnetic properties, which we cannot compare to theoretical values. Some improvements can be done even with the current results, however, this will not provide sufficient conclusions. I suggest to perform the magnetic measurements, when the situations with the SARS-CoV-2 will get better. Some other comments below.
- You can try to compare your data with a Brillouin function
- I do not agree with the sentence: “Transition from Mnq2 powders to Mnq2 nanorods doesn’t change the magnetic property of the sample.” The magnetization for the crystal sample is increasing more than two times faster. Why? There must be a reason.
- What is the unit of the calculated Curie constant? For example, one Mn(II) ion per mole of a compound gives the Curie constant around 4.38 emu K /mol. How to understand the presented result? Even with correctly calculated value of Curie constant, there is no data to compare.
- I agree that the most probably we see some kind of paramagnetic behavior. However, due to lack of other magnetic measurements we cannot even confirm that the chemical composition agrees with magnetic properties.
Reviewer 2 Report
I am sorry to let you know that the changes in the revised version are well done but not enough. I am still not convinced that powder and crystals are the same compound and that the only difference is that the powder has only some impurities.
Further analyzes must be performed like: ICP, TEM, IR and Elemental analysis, to confirm that. I know that because of the Covid-19 you could not perform these analyzes but it is better to publish later but rigorous results than faster but not in scientific way Many thanks for your comprehension.
